# GNFactor: Multi-Task Real Robot Learning with Generalizable Neural Feature Fields

**Yanjie Ze**[1*]   **Ge Yan**[2*]   **Yueh-Hua Wu**[2*]   **Annabella Macaluso**[2]

**Yuying Ge**[3]   **Jianglong Ye**[2]   **Nicklas Hansen**[2]   **Li Erran Li**[4]   **Xiaolong Wang**[2]

[1]Shanghai Jiao Tong University   [2]UC San Diego   [3]University of Hong Kong   [4]AWS AI, Amazon

[*]Equal Contribution

**yanjieze.com/GNFactor**

**Abstract:** It is a long-standing problem in robotics to develop agents capable of executing diverse manipulation tasks from visual observations in unstructured real-world environments. To achieve this goal, the robot needs to have a comprehensive understanding of the 3D structure and semantics of the scene. In this work, we present **GNFactor**, a visual behavior cloning agent for multi-task robotic manipulation with **G**eneralizable **N**eural feature **F**ields. GNFactor jointly optimizes a generalizable neural field (GNF) as a reconstruction module and a Perceiver Transformer as a decision-making module, leveraging a shared deep 3D voxel representation. To incorporate semantics in 3D, the reconstruction module utilizes a vision-language foundation model (*e.g.*, Stable Diffusion) to distill rich semantic information into the deep 3D voxel. We evaluate GNFactor on 3 real robot tasks and perform detailed ablations on 10 RLBench tasks with a limited number of demonstrations. We observe a substantial improvement of GNFactor over current state-of-the-art methods in seen and unseen tasks, demonstrating the strong generalization ability of GNFactor.

**Keywords:** Robotic Manipulation, Neural Radiance Field, Behavior Cloning

## 1   Introduction

One major goal of introducing learning into robotic manipulation is to enable the robot to effectively handle unseen objects and successfully tackle various tasks in new environments. In this paper, we focus on using imitation learning with a few demonstrations for multi-task manipulation. Using imitation learning helps avoid complex reward design and training can be directly conducted on the real robot without creating its digital twin in simulation [1, 2, 3, 4]. This enables policy learning on diverse tasks in complex environments, based on users' instructions (see Figure 1). However, working with a limited number of demonstrations presents great challenges in terms of generalization. Most of these challenges arise from the need to comprehend the 3D structure of the scene, understand the semantics and functionality of objects, and effectively follow task instructions based on visual cues. Therefore, a comprehensive and informative visual representation of the robot's observations serves as a crucial foundation for generalization.

The development of visual representation for robot learning has mainly focused on learning within a 2D plane. Self-supervised objectives are leveraged to pre-train the representation from the 2D image observation [6, 7, 8] or jointly optimized with the policy gradients [9, 10, 11]. While these approaches improve sample efficiency and lead to more robust policies, they are mostly applied to relatively simple manipulation tasks. To tackle more complex tasks requiring geometric understanding (*e.g.*, object shape and pose) and with occlusions, 3D visual representation learning has been recently adopted with robot learning [11, 12]. For example, Driess et al. [12] train the 3D scene representation by using NeRF and view synthesis to provide supervision. While it shows effectiveness over tasks requiring geometric reasoning such as hanging a cup, it only handles the simple

7th Conference on Robot Learning (CoRL 2023), Atlanta, USA.

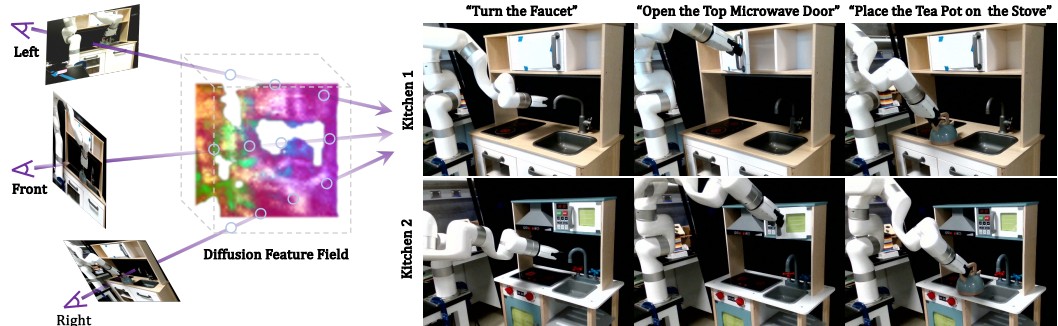

Figure 1: **Left:** Three camera views used in the real robot setup to reconstruct the feature field generated by Stable Diffusion [5]. We segment the foreground feature for better illustration. **Right:** Three language-conditioned real robot tasks across two different kitchens.

scene structure with heavy masking in a single-task setting. More importantly, without a semantic understanding of the scene, it would be very challenging for the robot to follow the user's language instructions.

In this paper, we introduce learning a language-conditioned policy using a novel representation leveraging both 3D and semantic information for multi-task manipulation. We train **G**eneralizable **N**eural **F**eature **F**ields (**GNF**) which distills pre-trained semantic features from 2D foundation models into the Neural Radiance Fields (NeRFs). We conduct policy learning upon this representation, leading to our model **GNFactor**. It is important to note that GNFactor learns an encoder to extract scene features in a feed-forward manner, instead of performing per-scene optimization in NeRF. Given a single RGB-D image observation, our model encodes it into a 3D semantic volumetric feature, which is then processed by a Perceiver Transformer [13] architecture for action prediction. To conduct multi-task learning, the Perceiver Transformer takes in language instructions to get task embedding, and reason the relations between the language and visual semantics for manipulation.

There are two branches of training in our framework (see Figure 3): (i) *GNF training*. Given the collected demonstrations, we train the Generalizable Neural Feature Fields using view synthesis with volumetric rendering. Besides rendering the RGB pixels, we also render the features of the foundation models in 2D space. The GNF learns from both pixel and feature reconstruction at the same time. To provide supervision for feature reconstruction, we apply a vision foundation model (*e.g.*, pre-trained Stable Diffusion model [5]) to extract the 2D feature from the input view as the ground truth. In this way, we can distill the semantic features into the 3D space in GNF. (ii) *GNFactor joint training.* Building on the 3D volumetric feature jointly optimized by the learning objectives of GNF, we conduct behavior cloning to train the whole model end-to-end.

For evaluation, we conduct real-robot experiments on three distinct tasks across two different kitchens (see Figure 1). We successfully train a single policy that effectively addresses these tasks in different scenes, yielding significant improvements over the baseline method PerAct [3]. We also conduct comprehensive evaluations using 10 RLBench simulated tasks [14] and 6 designed generalization tasks. We observe that GNFactor outperforms PerAct with an average improvement of 1.55x and 1.57x, consistent with the significant margin observed in the real-robot experiments.

## 2 Related Work

**Multi-Task Robotic Manipulation.** Recent works in multi-task robotic manipulation have led to significant progress in the execution of complex tasks and the ability to generalize to new scenarios [15, 2, 1, 16, 17, 3, 18, 19]. Notable methods often involve the use of extensive interaction data to train multi-task models [2, 1, 16, 17]. For example, RT-1 [1] underscores the benefits of task-agnostic training, demonstrating superior performance in real-world robotic tasks across a variety of datasets. To reduce the need for extensive demonstrations, methods that utilize keyframes – which encode the initiation of movement – have proven to be effective [20, 21, 22, 23, 24]. PerAct [3] employs the Perceiver Transformer [13] to encode language goals and voxel observations and shows its effectiveness in real robot experiments. In this work, we utilize the same action prediction frame-

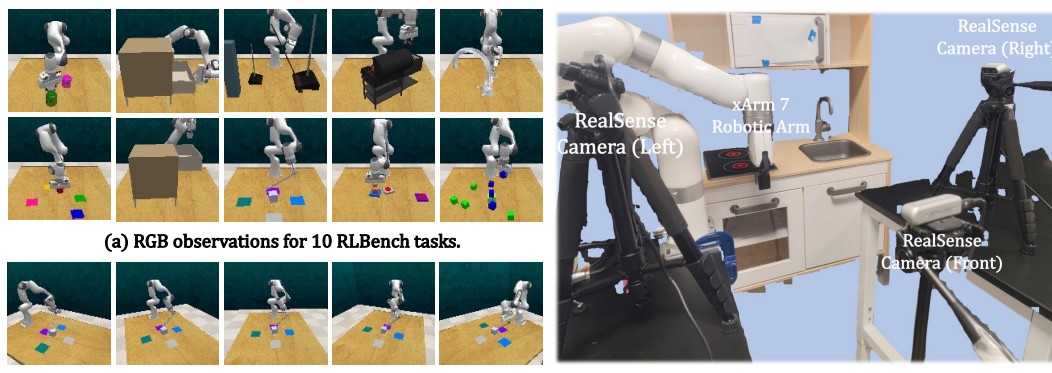

(a) RGB observations for 10 RLBench tasks.

(b) Sampled views for GNF training in simulation.

(c) Real robot setup.

Figure 2: **Simulation environments and the real robot setup.** We show the RGB observations for our 10 RLBench tasks in Figure (a), the sampled views for GNF in Figure (b), and the real robot setup in Figure (c).

work as PerAct while we focus on improving the generalization ability of this framework by learning a generalizable volumetric representation under limited data.

**3D Representations for Reinforcement/Imitation Learning (RL/IL).** To improve manipulation policies by leveraging visual information, numerous studies have concentrated on enhancing 2D visual representations [8, 7, 6, 25], while for addressing more complex tasks, the utilization of 3D representations becomes crucial. Ze et al. [11] incorporates a deep voxel-based 3D autoencoder in motor control, demonstrating improved sample efficiency compared to 2D representation learning methods. Driess et al. [12] proposes to first learn a state representation by NeRF and then use the frozen state for downstream RL tasks. While this work shows the initial success of utilizing NeRF in RL, its applicability in real-world scenarios is constrained due to various limitations: *e.g.*, the requirement of object masks, the absence of a robot arm, and the lack of scene structure. The work closest to ours is SNeRL [26], which also utilizes a vision foundation model in NeRF. However, similar to NeRF-RL [12], SNeRL masks the scene structure to ensure functionality and the requirement for object masks persists, posing challenges for its application in real robot scenarios. Our proposed GNFactor, instead, handles challenging muti-task real-world scenarios, demonstrating the potential for real robot applications.

**Neural Radiance Fields (NeRFs).** Neural fields have achieved great success in novel view synthesis and scene representation learning these years [27, 28, 29, 30, 31, 32], and recent works also start to incorporate neural fields into robotics [33, 34, 35, 12, 26]. NeRF [29] stands out for achieving photorealistic view synthesis by learning an implicit function of the scene, while it requires per-scene optimization and is thus hard to generalize. Many following methods [36, 37, 38, 39, 40, 41, 42] propose more generalizable NeRFs. PixelNeRF [43] and CodeNeRF [37] encode 2D images as the input of NeRFs, while TransINR [36] leverages a vision transformer to directly infer NeRF parameters. A line of recent works [44, 45, 46, 47, 48, 49] utilize pre-trained vision foundation models such as DINO [50] and CLIP [51] as supervision besides the RGB image, which thus enables the NeRF to learn generalizable features. In this work, we incorporate generalizable NeRF to reconstruct different views in RGB and embeddings from a pretrained Stable Diffusion model [5].

## 3   Method

In this section, we detail the proposed GNFactor, a multi-task agent with a 3D volumetric representation for real-world robotic manipulation. GNFactor is composed of a volumetric rendering module and a 3D policy module, sharing the same deep volumetric representation. The volumetric rendering module learns a Generalizable Neural Feature Field (GNF), to reconstruct the RGB image from cameras and the embedding from a vision-language foundation model, *e.g.*, Stable Diffusion [5]. The task-agnostic nature of the vision-language embedding enables the volumetric representation to learn generalizable features via neural rendering and thus helps the 3D policy module better handle multi-task robotic manipulation. The task description is encoded with CLIP [51] to obtain the task embedding $T$. An overview of GNFactor is shown in Figure 3.

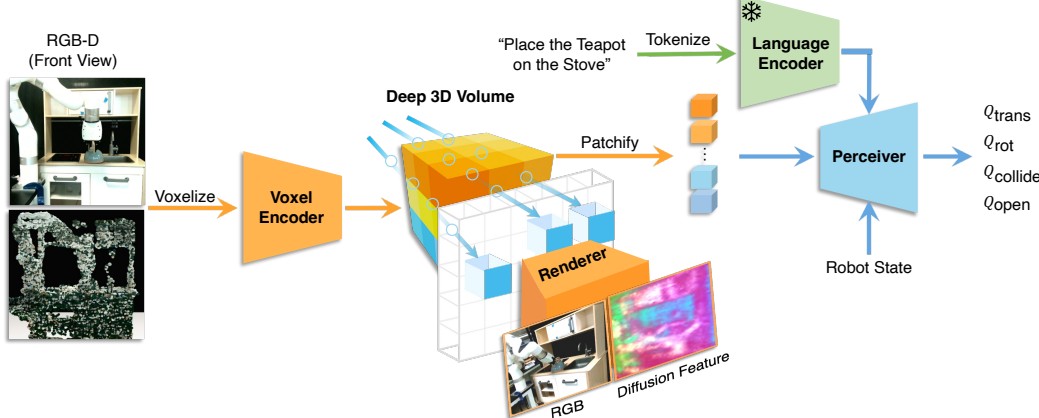

Figure 3: **Overview of GNFactor.** GNFactor takes an RGB-D image as input and encodes it using a voxel encoder to transform it into a feature in deep 3D volume. This volume is then shared by two modules: volumetric rendering (Renderer) and robot action prediction (Perceiver). These two modules are jointly trained, which optimizes the shared features to not only reconstruct vision-language embeddings (Diffusion Feature) and other views (RGB), but also to estimate accurate Q-values ($Q_{\text{trans}}$, $Q_{\text{rot}}$, $Q_{\text{collide}}$, $Q_{\text{open}}$).

### 3.1 Problem Definition

To effectively address complex real-world robotic problems, we structure the observation space as a 3D voxel space $\mathcal{O} \in \mathbb{R}^{100^3 \times 3}$, as opposed to the commonly used 2D images [1, 2, 7, 8]. The 3D voxel observation originates from an RGB-D image captured by a **single front camera** with known extrinsic and intrinsic parameters, ensuring our method's practical applicability in the real world. In addition to the front camera view used for policy training, we also gather additional $k$ views for training the GNF. We collect only RGB images for these additional views instead of RGB-D images. In real-world scenarios, we use $k = 2$, while in simulated environments, we set $k = 19$.

The action of the robot arm with a gripper is represented by translation $a_{\text{trans}} \in \mathbb{R}^3$, rotation $a_{\text{rot}} \in \mathbb{R}^{(360/5) \times 3}$, gripper openness $a_{\text{open}} \in [0, 1]$, and collision avoidance $a_{\text{collision}} \in [0, 1]$. For the rotation $a_{\text{rot}}$, each rotation axis is discretized into $R = 5$ bins. The collision avoidance parameter $a_{\text{collision}}$ instructs the motion planner regarding the necessity to avoid collisions, which is crucial as our tasks encompasses both contact-based and non-contact-based motions.

Due to the inefficiency of continuous action prediction and the extensive data requirements that come with it, we reformulate the behavior cloning problem as a *keyframe-prediction* problem [3, 52]. We first extract keyframes from expert demonstrations using the following metric: a frame in the trajectory is a keyframe when joint velocities approach zero and the gripper's open state remains constant. The model is then trained to predict the subsequent keyframe based on current observations. This formulation effectively transforms the continuous control problem into a discretized keyframe-prediction problem, delegating the internal procedures to the RRT-connect motion planner [53] in simulation and Linear motion planner in real-world xArm7 robot.

### 3.2 Learning Volumetric Representations with Generalizable Neural Feature Fields

In our initial step, we transform the RGB-D image into a $100^3$ voxel. Then the 3D voxel encoder encodes this 3D voxel and outputs our volumetric representation $v \in \mathbb{R}^{100^3 \times 128}$. To enhance the volumetric representation $v$ with structural knowledge and language semantics, we learn a Generalizable Neural Feature Field (GNF) that takes the deep volume $v$ as the scene representation and the model is learned by reconstructing the additional views and the features predicted by a 2D vision-language foundation model [5]. The entire neural rendering process is described as follows.

We denote $v_{\mathbf{x}} \in \mathbb{R}^{128}$ as the sampled 3D feature for the 3D point $\mathbf{x}$ using the volumetric representation $v$. $v_{\mathbf{x}}$ is formed with trilinear interpolation due to the discretized nature of the volume $v$. Our GNF primarily consists of three functions: (i) one density function $\sigma(\mathbf{x}, v_{\mathbf{x}}) : \mathbb{R}^{3+128} \mapsto \mathbb{R}_+$ that maps the 3D point $\mathbf{x}$ and the 3D feature $v_{\mathbf{x}}$ to the density $\sigma$, (ii) one RGB function $\mathbf{c}(\mathbf{x}, \mathbf{d}, v_{\mathbf{x}}) : \mathbb{R}^{3+3+128} \mapsto \mathbb{R}^3$ that maps the 3D point $\mathbf{x}$, the view direction $\mathbf{d}$, and the 3D feature $v_{\mathbf{x}}$ to color, and (iii) one vision-language embedding function $\mathbf{f}(\mathbf{x}, \mathbf{d}, v_{\mathbf{x}}) : \mathbb{R}^{3+3+128} \mapsto \mathbb{R}^{512}$ that maps the 3D

point $\mathbf{x}$, the view direction $\mathbf{d}$, and the 3D feature $v_{\mathbf{x}}$ to the vision-language embedding. In Figure 3, the corresponding components of these three functions are illustrated. Given a pixel's camera ray $\mathbf{r}(t) = \mathbf{o} + t\mathbf{d}$, which is defined by the camera origin $o \in \mathbb{R}^3$, view direction $\mathbf{d}$ and depth $t$ with bounds $[t_n, t_f]$, the estimated color and embedding of the ray can be calculated by:

$$\hat{\mathbf{C}}(\mathbf{r}, v) = \int_{t_n}^{t_f} T(t)\sigma(\mathbf{r}(t), v_{\mathbf{x}(t)})\mathbf{c}(\mathbf{r}(t), \mathbf{d}, v_{\mathbf{x}(t)})\mathrm{d}t\,,$$

$$\hat{\mathbf{F}}(\mathbf{r}, v) = \int_{t_n}^{t_f} T(t)\sigma(\mathbf{r}(t), v_{\mathbf{x}(t)})\mathbf{f}(\mathbf{r}(t), \mathbf{d}, v_{\mathbf{x}(t)})\mathrm{d}t\,, \tag{1}$$

where $T(t) = \exp\left(-\int_{t_n}^{t} \sigma(s)\mathrm{d}s\right)$. The integral is approximated with numerical quadrature in the implementation. Our GNF is then optimized to reconstruct the RGB image and the vision-language embedding from multiple views and diverse scenes by minimizing the following loss:

$$\mathcal{L}_{\mathrm{recon}} = \sum_{\mathbf{r} \in \mathcal{R}} \|\mathbf{C}(\mathbf{r}) - \hat{\mathbf{C}}(\mathbf{r})\|_2^2 + \lambda_{\mathrm{feat}}\|\mathbf{F}(\mathbf{r}) - \hat{\mathbf{F}}(\mathbf{r})\|_2^2\,, \tag{2}$$

where $\mathbf{C}(\mathbf{r})$ is the ground truth color, $\mathbf{F}(\mathbf{r})$ is the ground truth vision-language embedding generated by Stable Diffusion, $\mathcal{R}$ is the set of rays generated from camera poses, and $\lambda_{\mathrm{feat}}$ is the weight for the embedding reconstruction loss. For efficiency, we sample $b_{\mathrm{ray}}$ rays given one target view, instead of reconstructing the entire image. To help the GNF training, we use a coarse-to-fine hierarchical structure as the original NeRF [29] and apply depth-guided sampling [54] in the "fine" network.

## 3.3   Action Prediction with Volumetric Representations

The volumetric representation $v$ is optimized not only to achieve reconstruction of the GNF module, but also to predict the desired action for accomplishing manipulation tasks within the 3D policy. As such, we jointly train the representation $v$ to satisfy the objectives of both the GNF and the 3D policy module. In this section, we elaborate the training objective and the architecture of the 3D policy.

We employ a Perceiver Transformer [3] to handle the high-dimensional multi-modal input, *i.e.*, the 3D volume, the robot's proprioception, and the language feature. We first condense the shared volumetric representation $v$ into a volume of size $20^3 \times 128$ using a 3D convolution layer with a kernel size and stride of 5, followed by a ReLU function, and flatten the 3D volume into a sequence of small cubes of size $8000 \times 128$. The robot's proprioception is projected into a 128-dimensional space and concatenated with the volume sequence for each cube, resulting in a sequence of size $8000 \times 256$. We then project the language token features from CLIP into the same dimensions ($77 \times 256$) and concatenate these features with a combination of the 3D volume, the robot's proprioception state, and the CLIP token embedding. The result is a sequence with dimensions of $8077 \times 256$.

This sequence is combined with a learnable positional encoding and passed through the Perceiver Transformer, which outputs a sequence of the same size. We remove the last 77 features for the ease of voxelization [3] and reshape the sequence back to a voxel of size $20^3 \times 256$. This voxel is then upscaled to $100^3 \times 128$ with trilinear interpolation and referred to as $v_{\mathrm{PT}}$. $v_{\mathrm{PT}}$ is shared across three action prediction heads ($Q_{\mathrm{open}}$, $Q_{\mathrm{trans}}$, $Q_{\mathrm{rot}}$, $Q_{\mathrm{collide}}$ in Figure 3) to determine the final robot actions at the same scale as the observation space. To retain the learned features from GNF training, we create a skip connection between our volumetric representation $v$ and $v_{\mathrm{PT}}$. The combined volume feature $(v, v_{\mathrm{PT}})$ is used to predict a 3D Q-function $\mathcal{Q}_{\mathrm{trans}}$ for translation, as well as Q-functions for other robot operations like gripper openness ($\mathcal{Q}_{\mathrm{open}}$), rotation ($\mathcal{Q}_{\mathrm{rot}}$), and collision avoidance ($\mathcal{Q}_{\mathrm{collide}}$). The $\mathcal{Q}$-function here represents the action values of one timestep, differing from the traditional $\mathcal{Q}$-function in RL that is for multiple timesteps. For example, in each timestep, the 3D $\mathcal{Q}_{\mathrm{trans}}$-value would be equal to $1$ for the most possible next voxel and $0$ for other voxels. The model then optimizes the cross-entropy loss like a classifier,

$$\mathcal{L}_{\mathrm{action}} = -\mathbb{E}_{Y_{\mathrm{trans}}}\left[\log \mathcal{V}_{\mathrm{trans}}\right] - \mathbb{E}_{Y_{\mathrm{rot}}}\left[\log \mathcal{V}_{\mathrm{rot}}\right] - \mathbb{E}_{Y_{\mathrm{open}}}\left[\log \mathcal{V}_{\mathrm{open}}\right] - \mathbb{E}_{Y_{\mathrm{collide}}}\left[\log \mathcal{V}_{\mathrm{collide}}\right]\,, \tag{3}$$

where $\mathcal{V}_i = \mathrm{softmax}(\mathcal{Q}_i)$ for $\mathcal{Q}_i \in [\mathcal{Q}_{\mathrm{trans}}, \mathcal{Q}_{\mathrm{open}}, \mathcal{Q}_{\mathrm{rot}}, \mathcal{Q}_{\mathrm{collide}}]$ and $Y_i \in [Y_{\mathrm{trans}}, Y_{\mathrm{rot}}, Y_{\mathrm{open}}, Y_{\mathrm{collide}}]$ is the ground truth one-hot encoding. The overall learning objective for GNFactor is as follows:

$$\mathcal{L}_{\mathrm{GNFactor}} = \mathcal{L}_{\mathrm{action}} + \lambda_{\mathrm{recon}}\mathcal{L}_{\mathrm{recon}}\,, \tag{4}$$

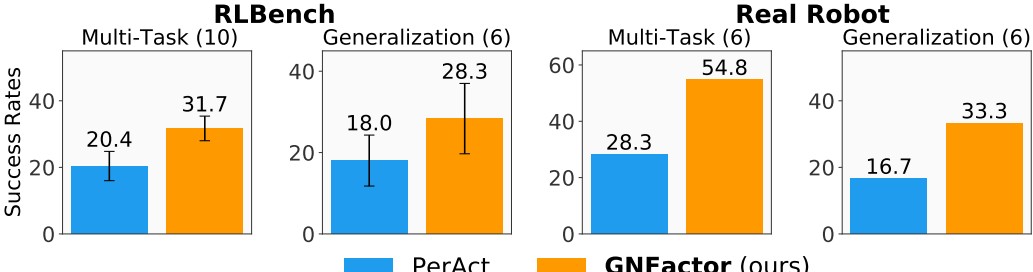

Figure 4: **Main experiment results.** We present the average success rates in both the multi-task and generalization settings across RLBench tasks and real robot tasks. The error bar represents one standard deviation. The number in the bracket denotes the number of tasks.

where $\lambda_{recon}$ is the weight for the reconstruction loss to balance the scale of different objectives. To train the GNFactor, we employ a joint training approach in which the GNF and 3D policy module are optimized jointly, without any pre-training. From our empirical observation, this approach allows for better fusion of information from the two modules when learning the shared features.

## 4 Experiments

In this section, we conduct experiments to answer the following questions: (i) Can GNFactor surpass the baseline model in simulated environments? (ii) Can GNFactor generalize to novel scenes in simulation? (iii) Does GNFactor learn a superior policy that handles real robot tasks in two different kitchens with noisy and limited real-world data? (iv) What are the crucial factors in GNFactor to ensure the functionality of the entire system? Our concluded results are given in Figure 4.

### 4.1 Experiment Setup

For the sake of reproducibility and benchmarking, we conduct our primary experiments in RLBench simulated tasks. Furthermore, to show the potential of GNFactor in the real world, we design a set of real robot experiments across two kitchens. We compare our GNFactor with the strong language-conditioned multi-task agent PerAct [3] in both simulation and the real world, emphasizing the universal functionality of GNFactor. Both GNFactor and PerAct use the single RGB-D image from the front camera as input to construct the voxel grid. In the multi-task simulation experiments, we also create a stronger version of PerAct by adding more camera views as input to fully cover the scene (visualized in Figure 10). Figure 2 shows our simulation tasks and the real robot setup. We briefly describe the tasks and details are left in Appendix B and Appendix C.

**Simulation.** We select 10 challenging language-conditioned manipulation tasks from the RLBench tasksuites [14]. Each task has at least two variations, totaling 166 variations. These variations encompass several types, such as variations in shape and color. Therefore, to achieve high success rates with very limited demonstrations, the agent needs to learn generalizable knowledge about manipulation instead of merely overfitting to the given demonstrations. We use the RGB-D image of size $128 \times 128 \times 3$ from the single front camera as the observation. To train the GNF, we also add additional 19 camera views to provide RGB images as supervision.

**Real robot.** We use the xArm7 robot with a parallel gripper in real robot experiments. We set up two toy kitchen environments to make the agent generalize manipulation skills across the scenes and designed three manipulation tasks, including *open the microwave door*, *turn the faucet*, and *relocate the teapot*, as shown in Figure 1. We set up three RealSense cameras around the robot. Among the three cameras, the front one captures the RGB-D observations for the policy training and the left/right one provides the RGB supervision for the GNF training.

**Expert Demonstrations.** We collect 20 demonstrations for each RLBench task with the motion planner. The task variation is uniformly sampled. We collect 5 demonstrations for each real robot task using a VR controller. Details for collection remain in Appendix D.

**Generalization tasks.** To further show the generalization ability of GNFactor, we design additional 6 simulated tasks and 3 real robot tasks based on the original training tasks and add task distractors.

**Training details.** One agent is trained with two NVIDIA RTX3090 GPU for 2 days (100k iterations) with a batch size of 2. The shared voxel encoder of GNFactor is implemented as a lightweight 3D UNet with only 0.3M parameters. The Perceiver Transformer keeps the same number of parameters as PerAct [3] (25.2M parameters), making our comparison with PerAct fair.

Table 1: **Multi-task test results on RLBench.** We evaluate 25 episodes for each checkpoint on 10 tasks across 3 seeds and report the success rates (%) of the final checkpoints. Our method outperforms the most competitive baseline PerAct [3] with an average improvement of $1.55$x and even still largely surpasses PerAct with 4 cameras as input. The additional camera views are visualized in Figure 10.

| Method / Task | close jar | open drawer | sweep to dustpan | meat off grill | turn tap | Average |
|---|---|---|---|---|---|---|
| PerAct | $18.7_{\pm8.2}$ | $54.7_{\pm18.6}$ | $0.0_{\pm0.0}$ | $40.0_{\pm17.0}$ | $38.7_{\pm6.8}$ | |
| PerAct (4 Cameras) | $21.3_{\pm7.5}$ | $44.0_{\pm11.3}$ | $0.0_{\pm0.0}$ | $65.3_{\pm13.2}$ | $46.7_{\pm3.8}$ | |
| GNFactor | $\mathbf{25.3}_{\pm6.8}$ | $\mathbf{76.0}_{\pm5.7}$ | $\mathbf{28.0}_{\pm15.0}$ | $57.3_{\pm18.9}$ | $\mathbf{50.7}_{\pm8.2}$ | |

| Method / Task | slide block | put in drawer | drag stick | push buttons | stack blocks | |
|---|---|---|---|---|---|---|
| PerAct | $18.7_{\pm13.6}$ | $2.7_{\pm3.3}$ | $5.3_{\pm5.0}$ | $18.7_{\pm12.4}$ | $\mathbf{6.7}_{\pm1.9}$ | 20.4 |
| PerAct (4 Cameras) | $16.0_{\pm14.2}$ | $\mathbf{6.7}_{\pm6.8}$ | $12.0_{\pm3.3}$ | $9.3_{\pm1.9}$ | $5.3_{\pm1.9}$ | 22.7 |
| GNFactor | $\mathbf{20.0}_{\pm15.0}$ | $0.0_{\pm0.0}$ | $\mathbf{37.3}_{\pm13.2}$ | $\mathbf{18.7}_{\pm10.0}$ | $4.0_{\pm3.3}$ | $\mathbf{31.7}$ |

Table 2: **Generalization to unseen tasks on RLBench.** We evaluate 20 episodes for each task with the final checkpoint across 3 seeds. We denote "L" as a larger object, "S" as a smaller object, "N" as a new position, and "D" as adding a distractor. Our method outperforms PerAct with an average improvement of $1.57$x.

| Method / Task | drag (D) | slide (L) | slide (S) | open (n) | turn (N) | push (D) | Average |
|---|---|---|---|---|---|---|---|
| PerAct | $6.6_{\pm4.7}$ | $\mathbf{33.3}_{\pm4.7}$ | $5.0_{\pm4.1}$ | $25.0_{\pm10.8}$ | $18.3_{\pm6.2}$ | $20.0_{\pm7.1}$ | 18.0 |
| GNFactor | $\mathbf{46.7}_{\pm30.6}$ | $25.0_{\pm4.1}$ | $\mathbf{6.7}_{\pm6.2}$ | $\mathbf{31.7}_{\pm6.2}$ | $\mathbf{28.3}_{\pm2.4}$ | $\mathbf{31.7}_{\pm2.4}$ | $\mathbf{28.3}$ |

## 4.2 Simulation Results

We report the success rates for multi-task tests on RLBench in Table 1 and for generalization to new environments in Table 2. We conclude our observations as follows:

**Dominance of GNFactor over PerAct for multi-task learning.** As shown by Table 1 and Figure 4, GNFactor achieves higher success rates across various tasks compared to PerAct, particularly excelling in challenging long-horizon tasks. For example, in sweep to dustpan task, the robot needs to first pick up the broom and use the broom to sweep the dust into the dustpan. We find that GNFactor achieves a success rate of $28.0\%$, while PerAct could not succeed at all. In simpler tasks like open drawer where the robot only pulls the drawer out, both GNFactor and PerAct perform reasonably well, with success rates of $76.0\%$ and $54.7\%$ respectively. Furthermore, we observe that enhancing PerAct with extra camera views does not result in significant improvements. This underscores the importance of efficiently utilizing the available camera views.

**Generalization ability of GNFactor to new tasks.** In Table 2, we observe that the change made on the environments such as distractors impacts all the agents negatively, while GNFactor shows better capability of generalization on 5 out of 6 tasks compared to PerAct. We also find that for some challenging variations such as the smaller block in the task slide (S), both GNFactor and PerAct struggle to handle. This further emphasizes the importance of robust generalization skills.

**Ablations.** We summarize the key components in GNFactor that contribute to the success of the volumetric representation in Table 4. From the ablation study, we gained several insights:

(i) Our GNF reconstruction module plays a crucial role in multi-task robot learning. Moreover, the RGB loss is essential for learning a consistent 3D feature in addition to the feature loss, especially since the features derived from foundation models are not inherently 3D consistent.

(ii) The volumetric representation benefits from Diffusion features and depth-guided sampling, where the depth prior is utilized to enhance the sampling quality in neural rendering. An intuitive explanation is that GNF, when combined with DGS, becomes more adept at learning depth and 3D structure information. This enhanced understanding allows the 3D representation to better concentrate on the surfaces of objects rather than the entire volume. Moreover, replacing Stable Diffusion with DINO [50] or CLIP [51] would not result in similar improvements easily, indicating the importance of our vision-language feature.

Table 4: **Ablations.** We report the averaged success rates on 10 RL-Bench tasks. "DGS" is short for depth-guided sampling. "$\rightarrow$" means replacing.

| Ablation | Success Rate (%) |
|---|---|
| GNFactor | **36.8** |
| w/o. GNF objective | 24.2 |
| w/o. RGB objective | 27.2 |
| w/o. Diffusion | 30.0 |
| Diffusion $\rightarrow$ DINO | 30.4 |
| Diffusion $\rightarrow$ CLIP | 32.0 |
| w/o. DGS | 29.2 |
| w/o. skip connection | 27.6 |
| $k = 19 \rightarrow 9$ | 33.2 |
| $\lambda_{\text{feat}} = 0.01 \rightarrow 1.0$ | 35.2 |
| $\lambda_{\text{recon}} = 0.01 \rightarrow 1.0$ | 35.2 |

Table 3: **Multi-task test results on real robot.** We evaluate 10 episodes for each task and report the resulting success rate (%). We denote "door" as "open door", "faucet" as "turn faucet", and "teapot" as "relocate teapot". The number in the parenthesis suggests the kitchen ID and "d" suggests testing with distractors.

| Method / Task | door (1) | faucet (1) | teapot (1) | door (1,d) | faucet (1,d) | teapot (1,d) | Average |
|---|---|---|---|---|---|---|---|
| PerAct | 30 | **80** | 0 | 10 | **50** | 0 | |
| GNFactor | **40** | **80** | **40** | **30** | **50** | **30** | |

| Method / Task | door (2) | faucet (2) | teapot (2) | door (2,d) | faucet (2,d) | teapot (2,d) | |
|---|---|---|---|---|---|---|---|
| PerAct | 10 | 50 | 0 | 10 | 30 | 0 | 22.5 |
| GNFactor | **50** | **70** | **40** | **20** | **40** | **30** | **43.3** |

(iii) While the use of skip connection is not a new story and we merely followed the structure of PerAct, the result of removing the skip connection suggests that our voxel representation, which distills features from the foundation model, plays a critical role in predicting the final action.

(iv) Striking a careful balance between the neural rendering loss and the action prediction loss is critical for optimal performance and utilizing information from multiple views by our GNF module proves to be beneficial for the single-view decision module.

Furthermore, we provide the view synthesis in the real world, generated by GNFactor in Figure 5 and Figure 6. We also give the quantitative evaluation measured by PSNR [29]. We observe that the rendered views are somewhat blurred since the volumetric presentation learned by GNFactor is optimized to minimize both the neural rendering loss as well as the action prediction loss, and the rendering quality is largely improved when the behavior cloning loss is removed and only the GNF is trained. Notably, for the view synthesis in the real world, we do not have access to ground-truth point clouds for either training or testing. Instead, the point clouds are sourced from RealSense cameras and are therefore imperfect. Despite the limitations in achieving accurate pixel-level reconstruction results, we focus on learning semantic understanding of the whole scene from distilling Diffusion features, which is more important for policy learning.

### 4.3 Real Robot Experiments

We summarize the results of our real robot experiment in Table 3. From the experiments, GNFactor outperforms the PerAct baseline on almost all tasks. Notably, in the *teapot* task where the agent is required to accurately determine the grasp location and handle the teapot from a correct angle, PerAct fails to accomplish the task and obtains a zero success rate across two kitchens. We observed that it is indeed challenging to learn a delicate policy from only 5 demonstrations. However, by incorporating the representation from the embedding of a vision-language model, GNFactor gains an understanding of objects. As such, GNFactor does not simply overfit to the given demonstrations.

The second kitchen (Figure 1) presents more challenges due to its smaller size compared to the first kitchen. This requires higher accuracy to manipulate the objects effectively. The performance gap between GNFactor and the baseline PerAct becomes more significant in the second kitchen. Importantly, our method does not suffer the same performance drop transitioning from the first kitchen to the second, unlike the baseline.

We also visualize our 3D policy module by Grad-CAM [55], as shown in Figure 7. We use the gradients and the 3D feature map from the 3D convolution layer after the Perceiver Transformer to compute Grad-CAM. We observe that the target objects are clearly attended by our policy, though the training signal is only the Q-value for a single voxel.

## 5 Conclusion and Limitations

In this work, we propose GNFactor, a visual behavior cloning agent for real-world multi-task robotic manipulation. GNFactor utilizes a Generalizable Neural Feature Field (GNF) to learn a 3D volumetric representation, which is also used by the action prediction module. We employ the vision-language feature from the foundation model Stable Diffusion besides the RGB feature to supervise the GNF training and observe that the volumetric representation enhanced by the GNF is helpful for decision-making. GNFactor achieves strong results in both simulation and the real world, across 10 RLBench tasks and 3 real robot tasks, showcasing the potential of GNFactor in real-world scenarios.

One major limitation of GNFactor is the requirement of multiple views for the GNF training, which can be challenging to scale up in the real world. Currently, we use three fixed cameras for GNFactor, but it would be interesting to explore using a cell phone to randomly collect camera views, where the estimation of the camera poses would be a challenge.

**Acknowledgment.** This work was supported, in part, by the Amazon Research Award, Cisco Faculty Award and gifts from Qualcomm.

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

# A   Visualizations

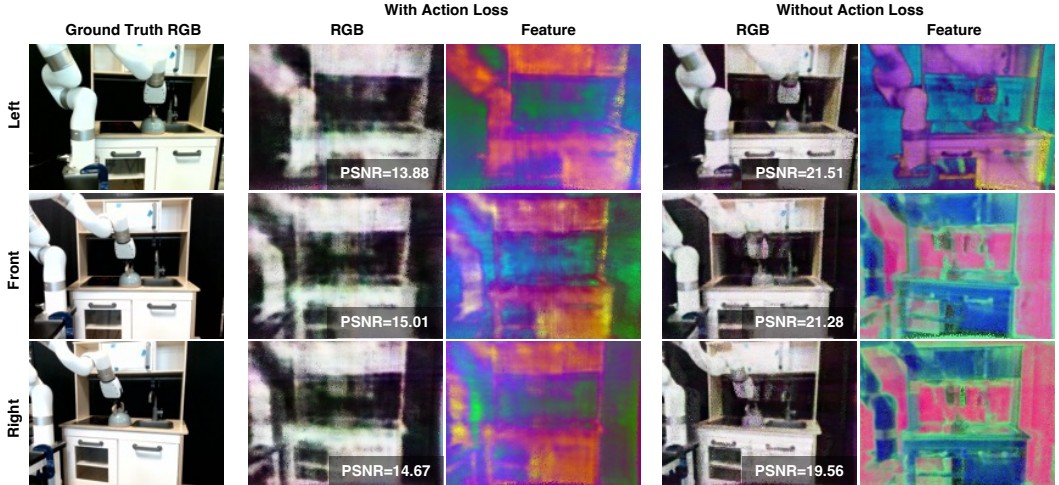

Figure 5: **View synthesis of GNFactor in the real world.** PSNR is computed for quantitative evaluation. The visualization with the action loss is relatively blurred compared to that without the action loss. The noisy rendering is mainly because, in inference, we do not optimize per-step for rendering but just perform one feedforward to obtain the feature.

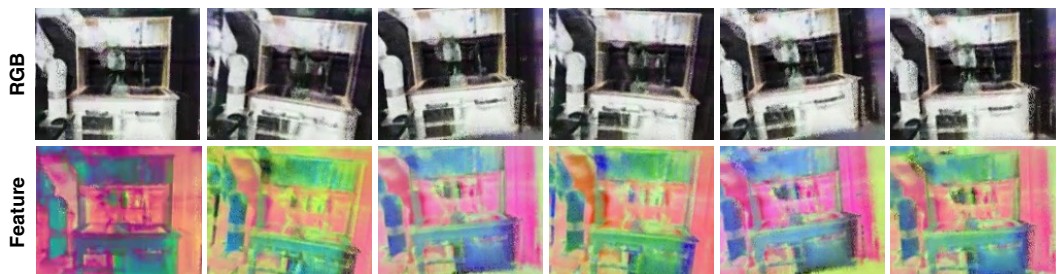

Figure 6: **More novel view synthesis results.** Both RGB and features are synthesized. We remove the action loss here for a better rendering quality. Videos are available on **yanjieze.com/GNFactor**.

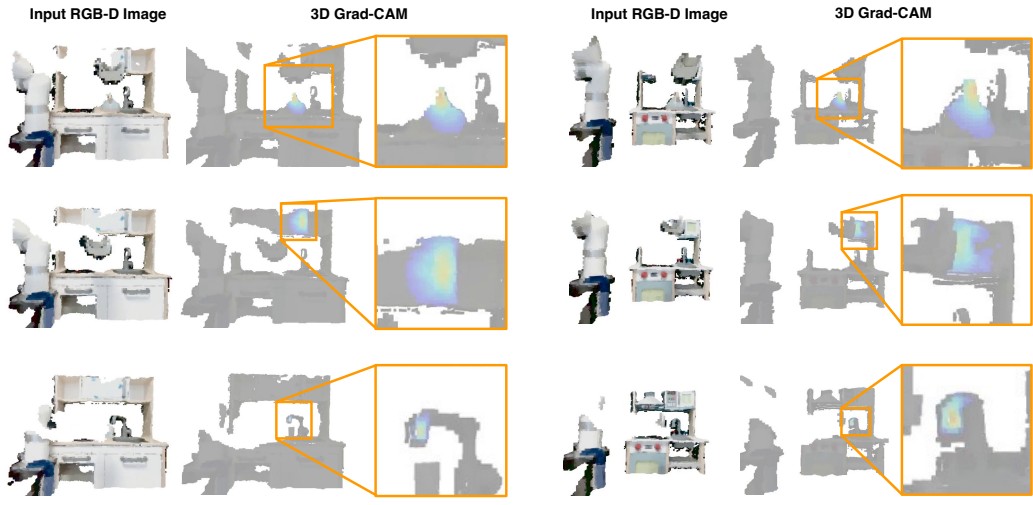

Figure 7: **Visualize the 3D policy module by Grad-CAM [55].** Though the supervision signal is only the Q-value for a single voxel during the training process, we observe in visualizations that the target objects are clearly attended by our policy. Videos are available on **yanjieze.com/GNFactor**.

## B    Task Descriptions

**Simulated tasks.** We select 10 language-conditioned tasks from RLBench [14], all of which involve at least 2 variations. See Table 5 for an overview. Our task variations include randomly sampled colors, sizes, counts, placements, and categories of objects, totaling 166 different variations. The set of colors have 20 instances: red, maroon, lime, green, blue, navy, yellow, cyan, magenta, silver, gray, orange, olive, purple, teal, azure, violet, rose, black, and white. The set of sizes includes 2 types: short and tall. The set of counts has 3 instances: 1, 2, 3. The placements and object categories are specific to each task. For example, `open drawer` has 3 placement locations: top, middle and bottom. In addition to these semantic variations, objects are placed on the tabletop at random poses within a limited range.

Table 5: **Language-conditioned tasks in RLBench [14].**

| Task | Variation Type | # of Variations | Avg. Keyframs | Language Template |
|------|----------------|-----------------|---------------|-------------------|
| close jar | color | 20 | 6.0 | "close the — jar" |
| open drawer | placement | 3 | 3.0 | "open the — drawer" |
| sweep to dustpan | size | 2 | 4.6 | "sweep dirt to the — dustpan" |
| meat off grill | category | 2 | 5.0 | "take the — off the grill" |
| turn tap | placement | 2 | 2.0 | "turn — tap" |
| slide block | color | 4 | 4.7 | "slide the block to — target" |
| put in drawer | placement | 3 | 12.0 | "put the item in the — drawer" |
| drag stick | color | 20 | 6.0 | "use the stick to drag the cube onto the — — target" |
| push buttons | color | 50 | 3.8 | "push the — button, [then the — button]" |
| stack blocks | color, count | 60 | 14.6 | "stack — — blocks" |

**Generalization tasks in simulation.** We design 6 additional tasks where the scene is changed based on the original training environment, to test the generalization ability of GNFactor. Table 6 gives an overview of these tasks. Videos are also available on **yanjieze.com/GNFactor**.

Table 6: **Generalization tasks based on RLBench.**

| Task | Base | Change |
|------|------|--------|
| drag (D) | drag stick | add two colorful buttons on the table |
| slide (L) | slide block | change the block size to a larger one |
| slide (S) | slide block | change the block size to a smaller one |
| open (n) | open drawer | change the position of the drawer |
| turn (N) | turn tap | change the position of the tap |
| push (D) | push buttons | add two colorful jar on the table |

**Real robot tasks.** In the experiments, we perform three tasks along with three additional tasks where distracting objects are present. The *door* task requires the agent to open the door on an mircowave, a task which poses challenges due to the precise coordination required. The *faucet* task requires the agent to rotate the faucet back to center position, which involves intricate motor control. Lastly, the *teapot* task requires the agent to locate the randomly placed teapot in the kitchen and move it on top of the stove with the correct pose. Among the three, the teapot task is considered the most challenging due to the random placement and the need for accurate location and rotation of the gripper. All 6 tasks are set up in two different kitchens, as visualized in Figure 8. The keyframes used in real robot tasks are given in Figure 9.

## C    Implementation Details

**Voxel encoder.** We use a lightweight 3D UNet (only 0.3M parameters) to encode the input voxel $100^3 \times 10$ (RGB features, coordinates, indices, and occupancy) into our deep 3D volumetric representation of size $100^3 \times 128$. Due to the cluttered output from directly printing the network, we

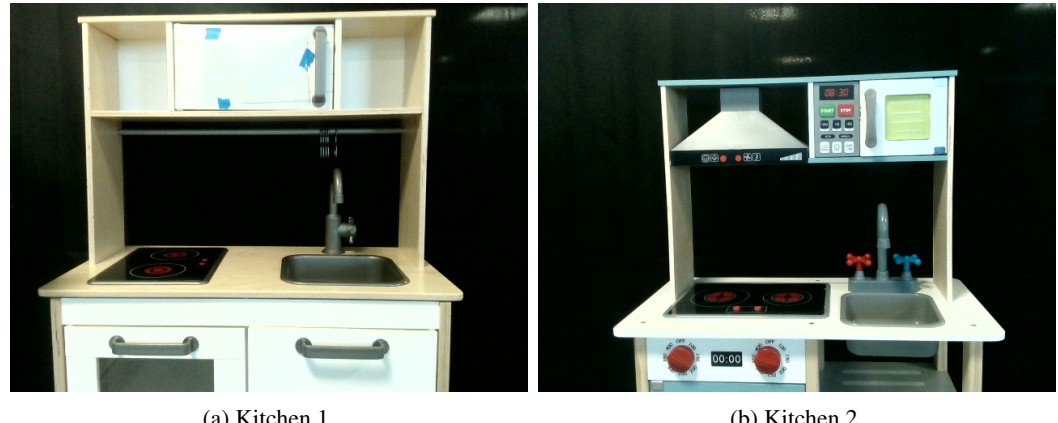

(a) Kitchen 1.  (b) Kitchen 2.

Figure 8: **Kitchens.** We give a closer view of our two kitchens for real robot experiments. The figures are captured in almost the same position to display the size difference between the two.

time ⟶

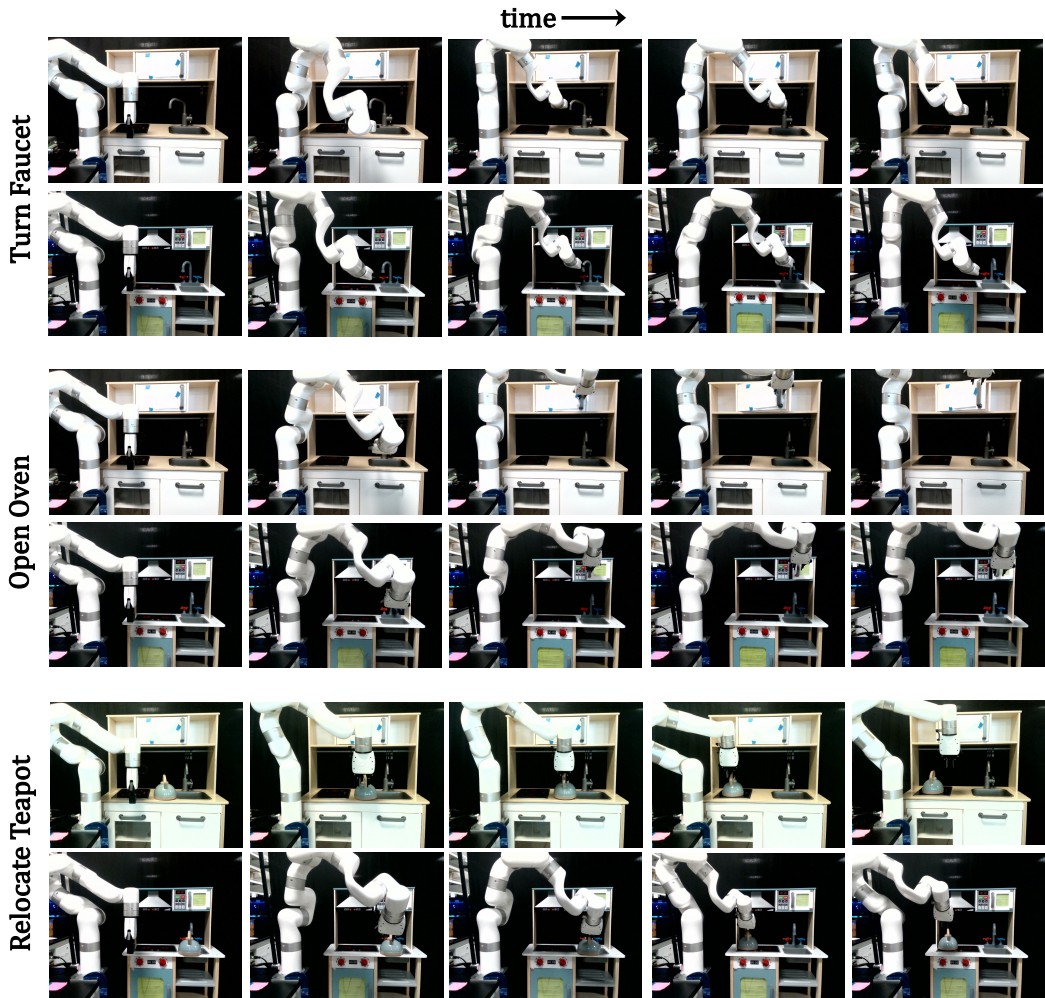

Figure 9: **Keyframes for real robot tasks.** We give the keyframes used in our 3 real robot tasks across 2 kitchens.

provide the PyTorch-Style pseudo-code for the forward process as follows. For each block, we use a cascading of one Convolutional Layer, one BatchNorm Layer, and one LeakyReLU layer, which is common practice in the vision community.

```
def forward(self, x):
    conv0 = self.conv0(x) # 100^3x8
    conv2 = self.conv2(self.conv1(conv0)) # 50^3x16
    conv4 = self.conv4(self.conv3(conv2)) # 25^3x32

    x = self.conv6(self.conv5(conv4)) # 13^3x64
    x = conv4 + self.conv7(x) # 25^3x32
    x = conv2 + self.conv9(x) # 50^3x16
    x = self.conv_out(conv0 + self.conv11(x)) # 100^3x128
    return x
```

**Generalizable Neural Field (GNF).** The overall network architecture of our GNF is close to the original NeRF [29] implementation. We use the same *positional encoding* as NeRF and the encoding function is formally

$$\gamma(p) = \left(\sin\left(2^0 \pi p\right), \cos\left(2^0 \pi p\right), \cdots, \sin\left(2^{L-1} \pi p\right), \cos\left(2^{L-1} \pi p\right)\right) . \tag{5}$$

This function is applied to each of the three coordinate values and we set $L = 6$ in our experiments. The overall position encoding is then 36-dimensional. The input of GNF is thus a concatenation of the original coordinates ($\mathbb{R}^3$), the position encoding ($\mathbb{R}^{36}$), the view directions ($\mathbb{R}^3$), and the voxel feature ($\mathbb{R}^{128}$), totaling 170 dimensions. Our GNF mainly consists of 5 `ResnetFCBlocks`, in which a skip connection is used. The input feature is first projected to 512 with a linear layer and fed into these blocks, and then projected to the output dimension 516 (RGB, density, and Diffusion feature) with a cascading of one ReLU function and one linear layer. We provide the PyTorch-Style pseudo-code for the networks as follows.

```
GNF(
  Linear(in_features=170, out_features=512, bias=True),
  (0-4): 5 x ResnetFCBlocks(
    (fc_0): Linear(in_features=512, out_features=512, bias=True)
    (fc_1): Linear(in_features=512, out_features=512, bias=True)
    (activation): ReLU()
  ),
  ReLU(),
  Linear(in_features=512, out_features=516, bias=True)
)
```

**Percevier Transformer.** Our usage of Percevier Transformer is close to PerAct [3]. We use 6 attention blocks to process the sequence from multi-modalities (3D volume, language token, and robot proprioception) and output a sequence also. The usage of Perceiver Transformer enables us to process the long sequence with computational efficiency, by only utilizing a small set of latents to attend the input. The output sequence is then reshaped back to a voxel to predict the robot action. The Q-function for translation is predicted by a 3D convolutional layer, and for the prediction of openness, collision avoidance, and rotation, we use global max pooling and spatial softmax operation to aggregate 3D volume features and project the resulting feature to the output dimension with a multi-layer perception. We could clarify that the design for the policy module is not our contribution; for more details please refer to PerAct [3] and its official implementation on https://github.com/peract/peract.

## D    Demonstration Collection for Real Robot Tasks

For the collection of real robot demonstrations, we utilize the HTC VIVE controller and basestation to track the 6-DOF poses of human hand movements. We then use triad-openvr package (https://github.com/TriadSemi/triad_openvr) to employ SteamVR and accurately map human operations onto the xArm robot, enabling it to interact with objects in the real kitchen.

We record the real-time pose of xArm and $640 \times 480$ RGB-D observations with the pyrealsense2 (https://pypi.org/project/pyrealsense2/). Though the image size is different from our simulation setup, we use the same shape of the input voxel, thus ensuring the same algorithm is used across the simulation and the real world. The downscaled images ($80 \times 60$) are used for neural rendering.

## E  Detailed Data

Besides reporting the final success rates in our main paper, we give the success rates for the best single checkpoint (*i.e.*, evaluating all saved checkpoints and selecting the one with the highest success rates), as shown in Table 7. Under this setting GNFactor outperforms PerAct with a larger margin. However, we do not use the best checkpoint in the main results for fairness.

We also give the detailed number of success in Table 8 for reference in addition to the success rates computed in Table 2.

Table 7: **Multi-task test results on RLBench.** We report the success rates for the best single checkpoint for reference. We could observe GNFactor surpasses PerAct by a large margin.

| Method / Task | close jar | open drawer | sweep to dustpan | meat off grill | turn tap | Average |
|---|---|---|---|---|---|---|
| PerAct | 22.7±5.0 | 62.7±13.2 | 0.0±0.0 | 46.7±14.7 | 36.0±9.8 | |
| GNFactor | **40.0**±**5.7** | **77.3**±**7.5** | **40.0**±**11.8** | **66.7**±**8.2** | **45.3**±**3.8** | |

| Method / Task | slide block | put in drawer | drag stick | push buttons | stack blocks | |
|---|---|---|---|---|---|---|
| PerAct | **22.7**±**6.8** | 9.3±5.0 | 12.0±6.5 | 18.7±6.8 | 5.3±1.9 | 23.6 |
| GNFactor | 18.7±10.5 | **10.7**±**12.4** | **73.3**±**13.6** | **20.0**±**3.3** | **8.0**±**0.0** | **40.0** |

Table 8: **Detailed data for generalization to novel tasks.** We evaluate 20 episodes, each across 3 seeds, for the final checkpoint and report the number of successful trajectories here.

| Generalization | PerAct | GNFactor w/o. Diffusion | GNFactor |
|---|---|---|---|
| drag (D) | $2, 0, 2$ | $15, 2, 5$ | $18, 5, 5$ |
| slide (L) | $6, 6, 8$ | $1, 10, 10$ | $6, 5, 4$ |
| slide (S) | $0, 2, 1$ | $6, 1, 5$ | $0, 3, 1$ |
| push (D) | $6, 3, 3$ | $4, 4, 5$ | $7, 6, 6$ |
| open (N) | $6, 2, 7$ | $5, 2, 9$ | $8, 5, 6$ |
| turn (N) | $4, 5, 2$ | $2, 7, 2$ | $6, 6, 5$ |

## F  Stronger Baseline

To make the comparison between our GNFactor and PerAct fairer, we enhance Peract's input by using 4 camera views, as visualized in Figure 10. These views ensure that the scene is fully covered. It is observed in our experiment results (Table 1) that GNFactor which takes the single view as input still outperforms PerAct with more views.

## G  Hyperparameters

We give the hyperparameters used in GNFactor as shown in Table 9. For the GNF training, we use a ray batch size $b_{\text{ray}} = 512$, corresponding to 512 pixels to reconstruct, and use $\lambda_{\text{feat}} = 0.01$ and $\lambda_{\text{recon}} = 0.01$ to maintain major focus on the action prediction. For real-world experiment, we set the weight of the reconstruction loss to 1.0 and the weight of action loss to 0.1. This choice was based on our observation that reducing the weight of the action loss and increasing the weight of the reconstruction loss did not significantly affect convergence but did help prevent overfitting to

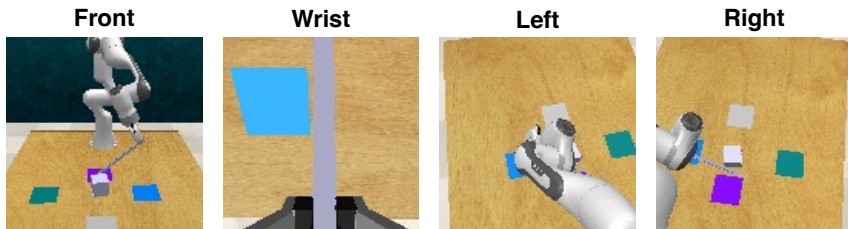

| Front | Wrist | Left | Right |

Figure 10: **Visualization of 4 cameras used for the stronger PerAct baseline.** To enhance the PerAct baseline, we add more views as the input of PerAct. These views are pre-defined in RLBench, making sure the observation covers the entire scene.

a limited number of real-world demonstrations. We uniformly sample 64 points along the ray for the "coarse" network and sample 32 points with depth-guided sampling and 32 points with uniform sampling for the "fine" network.

Table 9: **Hyperparameters** used in GNFactor.

| Variable Name | Value |
|:---:|:---:|
| training iteration | 100k |
| image size | $128 \times 128 \times 3$ |
| input voxel size | $100 \times 100 \times 100$ |
| batch size | 2 |
| optimizer | LAMB [56] |
| learning rate | 0.0005 |
| ray batch size $b_{\text{ray}}$ | 512 |
| weight for reconstruction loss $\lambda_{\text{recon}}$ | 0.01 |
| weight for embedding loss $\lambda_{\text{feat}}$ | 0.01 |
| number of transformer blocks | 6 |
| number of sampled points for GNF | 64 |
| number of latents in Perceiver Transformer | 2048 |
| dimension of Stable Diffusion features | 512 |
| dimension of CLIP language features | 512 |
| hidden dimension of NeRF blocks | 512 |

