# OpenReview forum: "GNFactor: Multi-Task Real Robot Learning with Generalizable Neural Feature Fields"
_robot-learning.org/CoRL/2023/Conference — CoRL 2023 Oral_

### Official Review · Reviewer_obdr · 2023-07-12

**Confidence:** 4
**Originality:** Very Good
**Technical Quality:** Very Good
**Clarity Of Presentation:** Excellent
**Impact:** 4

**Recommendation:**

Strong Accept: I recommend accepting the paper and will argue for my recommendation even if other reviewers hold a different opinion.

**Review:**

Strengths:
- Using NeRF rendering and diffusion consistency to improve the generalization performance of behavior-cloning agents is an interesting and novel approach. Prior works have either looked at 3D reconstruction (eg. MIRA) or pre-trained features (R3M) to improve policies, but GNFactor simply puts them together. This combination might be a great way of adapting internet pre-trained 2D models like CLIP/StableDiffusion to robotics tasks that involve 3D understanding.
- GNFactor achieves compelling quantitative results over PerAct (a prior state-of-the-art). The experiments are conducted in a benchmark simulation environment, and a code-release has been promised by the authors, so the results should be easily reproducible.
- Overall, the paper is well-written, and the supplementary materials are helpful. The figures are informative. The experiments cover a good range of tasks, ablations, random seed comparisons. The real-robot videos and simulation rollouts on the website and supplementary video are also nice.
- GNFactor works with real-robots. One major concern with NeRFs for robotics is the issue of painfully slow inference at test-time. Even the fastest NeRFs like Instant-NGP take a few seconds to optimize scenes. But GNFactor only uses the NeRF objective at train-time, and test-time inference is just a forward-pass.

Weaknesses:
-  The related work is missing a discussion on MIRA (Chen et al, 2021) and NeRF for dynamics (Li et al, 2020). The first work predicts affordances in rendered NeRF images. The second work uses NeRF reconstruction to enforce 3D persistence in visual dynamics models. While GNFactor uses NeRFs in a different context, a short discussion on these prior works might still be helpful to readers.
- Why are the rendered views in Figure 4 grainy and blurry? This quality is far worse than photorealistic renderings commonly found in NeRF papers. Section 4.2 briefly hints that this could be caused by “optimiz[ing] both neural rendering and action prediction loss”. Did the authors try training without the action prediction loss, and did that generate photorealistic renderings? It’s interesting that the NeRF loss still boosts task performance, despite the poor rendering quality. It might be worth investigating more deeply about this.
- In Section 4.2 (Ablations): did the authors try using CLIP vision features? Since the language is encoded with CLIP’s language encoder, wouldn’t CLIP vision features be better than Diffusion vision features since they are already aligned? If it turns out empirically that Diffusion features are indeed the best, then that would be good to know for readers.
- Minor: Table 1 - The average column could be merged between the two rows to avoid confusion.
- Minor: In Line 232 - “GNFactor achieves an impressive success rate of 28.0%” -> “impressive” is a bit of an overclaim for 28.0%

References:
Lin et al, 2022 – https://arxiv.org/abs/2212.06088
Li et al, 2021 –  https://arxiv.org/pdf/2107.04004.pdf

**Quality Of The Limitations Section:**

Limitations are addressed clearly

**Questions For Rebuttal:**

Summarizing questions from above:
- Why exactly are the rendered views grainy and blurry?
- Have the authors trained the NeRF without action loss?
- Have the authors tried using CLIP vision features instead of diffusion features?

**Robotics Focus:**

Sufficient demonstration on hardware

**Summary Of Paper:**

This paper presents GNFactor, a behavior-cloning agent that uses Neural Radiance Fields (NeRFs) to improve generalization with persistent 3D representations. GNFactor builds-off PerAct, a prior method for 3D manipulation, by adding an auxiliary NeRF objective that reconstructs RGB and Diffusion features from multiple viewpoints. This objective improves robustness and generalization by enforcing 3D and semantic consistency. Experiments are conducted in both simulation and real-world environments. In simulation, GNFactor significantly outperforms PerAct in task success, especially in tasks with unseen object instances of different sizes. In real-robot experiments, GNFactor outperforms PerAct by two-fold.


**Summary Of Recommendation:**

GNFactor proposes a novel objective to improve vision-based behavior-cloning policies. While prior works have investigated either 3D reconstruction or pre-trained vision features for better generalization, GNFactor combines them into a single objective. The experimental results and ablations indicate this objective makes a significant improvement. Overall, GNFactor is a valuable contribution to the manipulation community.

**Post Rebuttal**:
The authors clarified concerns, and conducted additional experiments to better understand the framework. Happy to raise the score to Strong Accept.

---

### Official Review · Reviewer_Lzg7 · 2023-07-13

**Confidence:** 4
**Originality:** Good
**Technical Quality:** Good
**Clarity Of Presentation:** Good
**Impact:** 3

**Recommendation:**

Weak Accept: I recommend accepting the paper, but will not argue for my recommendation if the majority of other reviewers have a different opinion.

**Review:**

# Summary
**Strengths**:
- The chosen combination of pre-existing tools proves to be effective in improving the performance in real-world and synthetic manipulation tasks, w.r.t. the baselines.
- The approach was tested on a real-world system.
- Videos and visualizations of the performed real-world and synthetic experiments are provided.
- The quality and clarity of the manuscript are overall good.

**Weaknesses**
- Additional experiments could be performed to further support the main claimed contribution that the improvement in generalization is a result of the used 3-D representation and the distilled features from a foundation model (cf. "Questions for Rebuttal")
- The system is fairly incremental.

--------
# Detailed review

## Originality and significance

The system is largely inspired by previous work (mainly PerAct) and is in this respect fairly incremental. However, the chosen combination of existing tools (language-conditioned Perceiver for action prediction, generalizable neural fields, and distillation of features from a foundational model) is interesting, and is shown to provide measurable improvements over previous works.

### Quality and clarity
- The manuscript is generally well-written, with only a limited amount of typos, although occasionally clarity, language and sentence structure could be improved (cf. below and "Questions for Rebuttal").
- The number of results presented, as well as the format of the presentation, is satisfactory, but additional ablations could be performed to strengthen the contribution (cf. "Questions for Rebuttal").
- The implementation details essential for reproducibility are presented (either in the main paper or in the Supplementary).

## Suggestions and minor questions
### Structure / clarity
- L13: "[...] substantial improvement": Improvement in what exactly?
- L25-26: It would be great if an example was provided for why "these challenges arise from the need to comprehend the 3D structure of the scene, understand the semantics and functionality of the objects [...]", etc.
- L37-38: "simple scene structure with heavy masking in a single-task setting". This sentence is unclear or not properly introduced. In particular: what does "masking" mean or refer to in this context?
- L88: The name NeRF-RL has not been introduced or linked to a reference before in the text.
- L93: Is [28] based on neural fields?
- L100: "the generalizable feature" is vague. Perhaps "generalizable features" would read better.
- L116: So far no mention of a "trajectory", "gripper" or a robot arm has been made and this terms might sound confusing to the reader. It might be beneficial to first state clearly that the problem involves a robot arm with a gripper. Similarly, some restructuring of Sec. 3.1 could be beneficial for understanding:
  - The camera configurations are introduced in L120-123, but frames are already mentioned in the previous paragraph.
  - L124-127 mention the collection of additional images to train the GNF, and in particular of RGB images rather than RGB-D ones, but it would be easier to follow this explanation if the training paradigm are requirements of each module were introduced first.
- L127: In what unit of measure is the translation regressed?
- L133: For additional clarity, I suggest pointing the reader here to the additional implementation details (e.g., the architecture of the encoder) introduced in App. B and Sec. 4.1.
- L146-147: The point "the transformation from the Deep 3D Volume to the 2D input for the Renderer corresponds to the density function" is not clear from Fig. 3, which in fact does not mention "density" at all.
- Eq. (2): Minor: The previously-introduced $v$ argument was dropped in $\hat{\mathbf{C}}(\mathbf{r}, v)$ and $\hat{\mathbf{F}}(\mathbf{r}, v)$.
- L167: How exactly is "the shared volumetric representation" "condensed" into a lower dimensional volume? Through some form of pooling?
- L169: What is meant exactly by "robot's proprioception"? That is, what does the robot state consist of exactly?
- L178: Why is $Q_\textrm{open}$ not mentioned here?
- Tables 1, 2, and 3 should include either in the number or in the caption the percent sign, since (I'm assuming) the success rate is expressed as percentage values here.
- While understandable, the symbols in Table 4 (particularly $\rightarrow$) could be clarified.

### Additional references
Additional references that distill pre-trained foundation models into 3-D representations could be included:
- Peng et al. "OpenScene 3D Scene Understanding with Open Vocabularies", CVPR, 2023.
- Murthy Jatavallabhula et al., "ConceptFusion: Open-set Multimodal 3D Mapping", RSS, 2023.
- Blomqvist et al., "Neural Implicit Vision-Language Feature Fields", arXiv/2303.10962, 2023.

### Language
- L9-10: Repetition of the same concept: "To incorporate semantics in 3D[....], to distill rich semantics into the deep 3D voxel".
- Tense coherency: In L39 and L47-50, the future tense is used, unlike the rest of the paragraph, where the present tense is used.
- L50: "reason the relations" is not correct English.
- L93-95: This sentence is not cohesive/contains a repetition: "NeRF[30] stands out [...], while NeRF requires [...]".
- L99: "which thus enabling" is not correct English.
- Caption of Table 1: "finally saved checkpoints" is a bit clunky. Perhaps "final checkpoint" would sound better.
- L239: "handle". Missing: handle what?
- L250: For coherency with the other occurencies, "the GNFactor" should be "GNFactor".

### Typos and formatting
- "e.g." -> "\eg" (L10, L56).
- "ground-truth" (noun) -> "ground truth" (L57). "ground truth" (adjective) -> "ground-truth" (L258)
- L72-73: em-dashes (--) should be used instead of simple dashes.

**Quality Of The Limitations Section:**

Limitations are addressed clearly

**Questions For Rebuttal:**

- The main point that I would like the authors to further support is the claim that the generalization ability comes from the use of a 3D representation and of features distilled from a foundation model, or in other words, from the multi-task training paradigm. As noted in L252-255, the rendering reconstructions are quite poor in quality, and by looking at Table 4 it seems like increasing the importance of the rendering losses results in a decreased success rate? Was a baseline with $\lambda_\textrm{recon}=0$ or at least with $\lambda_\textrm{feat}=0$ tried? Such an ablation would really show the importance of the multi-task training.
- For the same reason, it would be great if Table 3 included variances like Table 1 and 2, to show how large the improvement of generalization is w.r.t. PerAct.
- L86: Why is the "lack of scene structure" a limitation? What exactly is meant by "scene structure in this context"? Similarly, the sentence "SNeRL masks the scene structure to ensure functionality" is unclear: What is meant by "masking", "scene structure", and "ensuring functionality" (of what?) in this context?
- How is the coordinate frame of the voxel grid chosen? In particular, how are the encoded camera-centric features backprojected into the voxel volume?
- How important is it that the novel-view synthesis during training is performed for keyframes defined according to the description L115-L117? Would the results change significantly if less constrained viewpoints (e.g.,  with a varied gripper's open state, or with non-zero joint velocity) were used?
- Could you elaborate on the need to go back to a voxel representation (L176)? I see that this is practically used for the skip connection, but was the impact of this evaluated? For instance, how important is the skip connection?
- L257: What is meant by "components"?
- L269-270: "GNFactor gains an understanding of objects" sounds like an unsupported claim. I encourage the authors to rephrase or the further support this claim.

**Robotics Focus:**

Sufficient demonstration on hardware

**Summary Of Paper:**

The paper presents an approach for end-to-end multi-task robotic manipulation based on a language prompt. At test time, the system: a) encodes a third-view RGB-D image of the robot environment into a 3D-voxel-based volume and further takes as input: b) the robot state, c) a text-based description of a task to perform. Combining a), b), c), the system predicts a set of actions that the robot arm should execute. The same inputs are provided during training, which follows a multi-task learning paradigm: 1. Coordinate-based MLPs are conditioned on the voxel-based feature grid a) and decoded using volume rendering to reconstruct both RGB and foundation-model features of a novel view. 2. a), b) and c) are combined to estimate Q-values for the actions to perform. The approach outperforms previous methods in real-world and synthetic tasks. The main claimed contribution consists in generalization improvement, attributed to the used 3D representation and to the features from a foundation model.

**Summary Of Recommendation:**

While partially incremental, the proposed approach effectively combines tools, resulting in a performance improvement over previous works.
Considering also that the method was tested on a real-world system, overall I recommend the paper for acceptance. However, I would like the authors to clarify the points I raised above, and in particular to further support the claim that the improvement in generalization is a result of the use of a 3-D representation and of the features distilled from a foundation model.

---

> ### Author Response · Authors · 2023-08-14
> **Thank you for your comments**
>
> Dear reviewer,
>
> Since tomorrow is the deadline for the rebuttal period, we wonder if our replies have addressed your concerns? Please let us know if you have any other questions.
>
> Best,
>
> Paper50 authors

---

> > ### Comment · Reviewer_Lzg7 · 2023-08-14
> > **Response to the Authors**
> >
> > Yes, thank you. I confirm my rating.

---

> > > ### Author Response · Authors · 2023-08-15
> > > **Thank you!**
> > >
> > > Dear reviewer,
> > >
> > > We are glad that all of your concerns have been addressed. We are wondering if would consider increase your rating to strong accept, or are there still concerns preventing you from giving the strong accept.
> > >
> > > Best,
> > >
> > > Authors

---

### Official Review · Reviewer_dBGA · 2023-07-13

**Confidence:** 5
**Originality:** Excellent
**Technical Quality:** Very Good
**Clarity Of Presentation:** Good
**Impact:** 4

**Recommendation:**

Weak Accept: I recommend accepting the paper, but will not argue for my recommendation if the majority of other reviewers have a different opinion.

**Review:**

The biggest strength and main contribution of this paper is the way in which the voxel encoder is trained. In addition to a loss on the output Q-values, a secondary loss on the reconstruction of the scene is used. Specifically, the encoded voxel grid is interpreted as a neural radiance field and is re-rendered from another viewpoint using the NeRF volumetric rendering equation. The loss between this image and the ground truth RGB image taken from this viewpoint is used to condition the voxel encoder. Additionally, a feature image is rendered from the same viewpoint using the same technique and compared to the features generated by Stable Diffusion for that image. These two extra loss terms enforce that the voxel encoder encodes both correct geometry and meaningful semantics, significantly improving its performance as shown in the results.

The main weakness of the paper is the results are not very persuasive. The goal is to show that the robot can generalize to new tasks not seen during training using this method, but the average generalization success rate is only around 30%. It’s hard to say if the robot is really succeeding or not. However, the inclusion of the PerAct baseline goes a long way to alleviating this concern. It is clear that the methodology in this paper is doing something because the baseline is even worse at generalization. Without this baseline that would be impossible to tell. One improvement the authors might consider to this paper would be to add additional baselines, particularly ones that vary significantly or use alternative kinds of inputs. This would help put in context the performance of the proposed method. Although I will note that even without additional baselines, I would recommend this paper for acceptance, so they aren’t strictly necessary, more of a nice finishing touch.


**Quality Of The Limitations Section:**

Limitations are addressed clearly

**Questions For Rebuttal:**

The biggest area for improvement during the rebuttal period is clarity. There are some things in the paper I am unclear about. For example, in section 3.3 it says on line 167 that the voxel grid is “condensed” and then on line 177 it is “upscaled.” How? Please clarify which method is used for these (a couple words is likely sufficient, e.g., “using trilinear downsampling”). On a bigger scale, it is unclear how the robot actually executes actions. Section 3.1 mentions keyframes and using a motion planner between them, but which planner? Furthermore, the perceiver part of the network outputs what the paper terms “Q-values.” Are these Q-values as in Q-learning used by many RL methods? Or are these the next desired kinematic state for the planner to plan to? If the latter, please pick a different variable than “Q” as this will be very confusing to anyone from the RL field, and if the former please describe in much more detail how actions are selected. I suspect the latter is the case but the choice of variable name confused me for quite some time.

Small nit: The sentence spanning lines 233 and 234 lists “GNFactor and PerAct” in that order but lists the results “54.7% and 76.0%” in the opposite order, with PerAct achieving 54.7% and GNFactor achieving 76.0% (at least according to Table 1). Please swap.


**Robotics Focus:**

Sufficient demonstration on hardware

**Summary Of Paper:**

This paper proposes a method for performing language-conditioned actions across multiple tasks using a neural feature field representation. The robot is given a natural language description of the task and expected to accomplish it. To do this, first an RGBD image from the robot’s perspective is captured and converted to a voxel grid. Next it is fed to a learned voxel encoder which outputs a voxel feature grid. This is then condensed and concatenated with both proprioceptive and CLIP features from the natural language description of the task. The result of this is fed through the perceiver network, which outputs Q-values for the robot’s actions. The results show that this method outperforms the baseline on both simulated and real robot tasks.

**Summary Of Recommendation:**

The methodology proposed in this paper is novel and interesting. The results clearly show that it improves over the chosen baseline. As such, I think this paper makes a meaningful contribution to the literature. So long as the authors can address my concerns about the clarity of the paper, I would give this paper a weak acceptance recommendation. If the authors are able to add additional baseline comparisons, I would bump that up to a strong acceptance recommendation.

---

> ### Author Response · Authors · 2023-08-14
> **Thank you for your comments**
>
> Dear reviewer,
>
> Since tomorrow is the deadline for the rebuttal period, we wonder if our replies have addressed your concerns? Please let us know if you have any other questions.
>
> Best,
>
> Paper50 authors

---

> > ### Comment · Reviewer_dBGA · 2023-08-14
> > **Work-life balance**
> >
> > Apologies for taking so long, as a rule I don't work on weekends and unfortunately the organizers decided to have the rebuttal period include one. I've responded to your rebuttal below.

---

### Official Review · Reviewer_asyz · 2023-07-20

**Confidence:** 3
**Originality:** Good
**Technical Quality:** Good
**Clarity Of Presentation:** Good
**Impact:** 3

**Recommendation:**

Weak Accept: I recommend accepting the paper, but will not argue for my recommendation if the majority of other reviewers have a different opinion.

**Review:**

Strengths:


- The paper is well-organized and easy to follow
- Experiments in both simulation and real-world are conducted
- The proposed methods outperform the baseline in empirical results

Weakness

- The view synthesis results are poor, and it's unclear if GNF learns visual features. The author may consider ablative studies on the RGB reconstruction module (e.g. only remove RGB reconstruction loss and keep the semantic feature reconstruction loss). Also, the authors may need to provide more details on how the GNF is implemented to capture high-frequency geometry and texture (e.g. details on positional encoding), as the local features $v_x$ lack high-frequency information (since it's an interpolation of a discretized feature grid)

- The proposed GNF seems to be different from NeRF, as NeRF is an implicit field that takes position (positional encoder) as input, while GNF uses both position and local features (interpolation of a feature grid), which is more similar to GIGA [1]. Only the volume rendering part shares similarity with NeRF. The author may explain more about why they are not using the original NeRF architecture.

- More details on how PerAct is trained in the experiments section. Is the voxel grid constructed from multiple RGB-D images or just one RGB-D image (as in GNFactor)? Using only one RGB-D camera to train PerAct may lead to an unfair comparison, as GNFactor utilizes additional RGB images, gaining more information regarding the scene during training. While PerAct may learn poorly in certain scenarios due to limited data, such as when the target object is occluded in the RGB-D image.

**Quality Of The Limitations Section:**

Limitations are addressed clearly

**Questions For Rebuttal:**

- Why the input feature of GNF is 170 in L460, it seems to be inconsistent with the definition in L141-L144. I'm also confused why the input of σ,c,f is $R^{3}$, $R^{3 \times 3}$, $R^{3 \times 3}$ respectively. Also, what is the size of the positional encoding?
- More explanation on ablation results, why Diffusion works better than DINO?
- Why might depth-guided sampling (DGS) have a more significant impact on the model's performance even than semantic features, considering its originally proposed for speeding up rendering, to my understanding?
- I would suggest the author avoid using the term "NeRF" to describe GNF (e.g. in L7) to prevent confusion (see weakness)
- Missing reference [1]

Reference:

[1] Z. Jiang et al., GIGA: "Synergies Between Affordance and Geometry: 6-DoF Grasp Detection via Implicit Representations", RSS, 2021

**Robotics Focus:**

Sufficient demonstration on hardware

**Summary Of Paper:**

This paper proposed an improved version of PerAct for better generalization ability. Specifically, The author proposed Generalizable Neural feature Fields (GNF) as a visual representation of the scene, and combine it with the perceiver transformer-based action module in PerAct. RGB and semantic feature reconstruction via volume rendering is used to aid in learning the GNF.

**Summary Of Recommendation:**

Based on the strengths and weaknesses section, the paper's empirical results and usage of semantic feature reconstruction are interesting, I would recommend weak accept for the paper.

---

### Author Response · Authors · 2023-08-13
**Summary of the Rebuttal**

We thank all the reviewers for your insightful comments. Based on your feedback, we have provided the following new results:

- Provide better view synthesis results ([link](https://drive.google.com/file/d/1d1cyhEYEDT2u_pWaS4BAUNXBCQZjqbh8/view?usp=sharing)) by training the GNF only without the behavior cloning loss.
- Add a stronger baseline by enhancing PerAct with additional camera views as input, for a more fair comparison. The input camera views are visualized [here](https://drive.google.com/file/d/1U_ZLEKWr2BwU5Ci-XWppbmmTdeiQssKm/view?usp=sharing) and results are provided in the [table](https://drive.google.com/file/d/1Emmj5OJKBssnG5ip9pgC49oToIdl70il/view?usp=sharing). It is shown that even though the input views of PerAct fully cover the scene, our method could still outperform it by a large margin, emphasizing the importance of our GNF module.
- Add ablation studies on GNFactor across 10 RLBench tasks to validate the effectiveness of our approach including: (i) training without GNF objectives; (ii) training without RGB reconstruction; (iii) training without skip connection; (iv) replacing Stable Diffusion with CLIP in training.

Additionally, we would carefully revise our manuscripts based on the reviewers’ suggestion.

Again, we thank all the reviewers for your constructive feedbacks. We believe that all comments have been addressed, but are happy to address any further comments from reviewers.

Best,

Authors of Paper 50

---

### Decision · Program_Chairs · 2023-08-30

**Decision:**

Accept (Oral)

**Comment:**

The paper introduces GNFactor, a robotic agent designed for multi-task manipulations based on visual observations in unstructured real-world scenarios. GNFactor combines a neural radiance field (NeRF) for scene reconstruction and a Perceiver Transformer for decision-making, using a unified deep 3D voxel representation. To infuse semantics in 3D, GNFactor integrates a vision-language foundation model like Stable Diffusion. Evaluations on a real robot and RLBench reveal that GNFactor significantly outperforms existing techniques, showcasing strong generalization capabilities.

A common consensus from the reviewers indicates that while the success rate isn't necessarily groundbreaking, the innovative approach, comparison with baselines, and evidence of general improvement over previous methods make the work a valuable contribution to the field.

More specifically, the reviewers unanimously acknowledged the structure and clarity of the paper, the evaluations in both simulated and real-world environments, and the superior performance over the baselines in empirical results. The reviewers also acknowledged the novelty of combining NeRF rendering and diffusion consistency to enhance the generalization performance of behavior-cloning agents.

Given the rebuttal from the authors and the discussions with the reviewers, I recommend accepting this paper. However, the authors shall revise the paper based on the reviewers' comments by integrating the suggested improvements and responses to the reviewers' concerns.